# Investigation of Flame Retardant Flexible Polyurethane Foams Containing DOPO Immobilized Titanium Dioxide Nanoparticles

**DOI:** 10.3390/polym11010075

**Published:** 2019-01-05

**Authors:** Quanxiao Dong, Keyu Chen, Xiaodong Jin, Shibing Sun, Yingliang Tian, Feng Wang, Peng Liu, Mingshu Yang

**Affiliations:** 1College of Materials Science and Engineering, Beijing University of Technology, Beijing 100024, China; quanxiao185@iccas.ac.cn (Q.D.); summercky@126.com (K.C.); jinxiaodong@bjut.edu.cn (X.J.); tianyl@bjut.edu.cn (Y.T.); 2Railway Engineering Research Institute, China Academy of Railway Sciences, Beijing 100081, China; 3National Laboratory for Molecular Science, CAS Key Laboratory of Engineering Plastics, Institute of Chemistry, Chinese Academy of Sciences, Beijing 100190, China; liupeng@iccas.ac.cn (P.L.); yms@iccas.ac.cn (M.Y.)

**Keywords:** titanium dioxide, polyurethane, surface modification, supercritical carbon dioxide, flame retardant

## Abstract

In this work, a multi-functional nanoparticle (TiO_2_-KH570-DOPO) has been successfully synthesized through the attachment of 9,10-dihydro-9-oxa-10-phosphaphenanthrene-10-oxide (DOPO)-methacryloxy propyl trimethoxyl silane on the surface of titanium dioxide (TiO_2_). Supercritical carbon dioxide was used as the solvent in order to increase the grafting level. The chemical structure of TiO_2_-KH570-DOPO was fully characterized using Fourier transform infrared spectra, thermogravimetric analysis and transmission electron microscopy. The modified TiO_2_ was incorporated into flexible polyurethane foam (FPUF). The fire performance of FPUF blends was evaluated using microscale combustion calorimetry. Peak heat release rate and total heat release values were reduced from 657.0 W/g and 28.9 kJ/g for neat FPUF sample to 519.2 W/g and 26.8 kJ/g of FPUF specimen containing 10 wt % of TiO_2_-KH570-DOPO. Analysis of thermal stability and the observation of char formation suggests that TiO_2_-KH570-DOPO is active in the condensed phase.

## 1. Introduction

Due to high resilience and low density, flexible polyurethane foams (FPUFs) are widely used in everyday items, such as vehicles and furniture [1,2,3,4,5,6]. However, the inherent structure of FPUF (large surface area and good air permeability) makes it highly flammable [5], which presents a threat to both the performance and durability of products containing this material. Therefore, developing effective flame retardant FPUPs is urgently needed.

The incorporation of flame retardants has been widely applied to enhance the fire resistance of FPUFs. Halogen-containing, phosphorus-based, intumescent flame retardants (IFR), metal hydroxides and nanoparticles [7,8,9] have all been used to improve the fire safety of polyurethane materials. Halogen-containing flame retardants have gradually been replaced by halogen-free flame retardants due to increased environmental concerns and potential negative health effects of these materials. However, high loading of inorganic halogen-free flame retardants (such as metal hydroxides) obviously often degrades the mechanical performance of the polymer matrix. Recently, the surface treatments of FPUF through layer-by-layer, plasma and sol-gel techniques have received increasing attention due to high efficiency and rapid preparation [10]. Branched polyethylenimine (BPEI), chitosan and poly(acrylic acid) (PAA) have all been used to improve the flame retardant performance of PU foams [11]. However, the durability of surface coatings remains a huge challenge. Washing, chemical etching and physical abrasion all degrade the surface coating [12].

Incorporation of titanium dioxide (TiO_2_) exerts positive effects on the charring process for polymer matrices undergoing combustion [13,14,15]. A problem is that TiO_2_ particles do not disperse well within most polymer matrices because of the polarity differences. As a result, surface modification with silicanes, such as γ-aminopropyl triethoxysilane (KH550) and methacryloxy propyl trimethoxyl silane (KH570), is needed for incorporation of TiO_2_. 9,10-dihydro-9-oxa-10-phosphaphenanthrene-10-oxide (DOPO) has been proven to be an effective flame retardant in both the gas and condensed phase [16,17,18]. The presence of KH570 facilitates the dispersion of the flame retardant in the polymer matrix [19,20]. In previous work, KH570-DOPO was successfully prepared and applied to increase the fire resistance of polypropylene [21,22]. However, the grafting level was not satisfactory, which might be the reason for the low efficiency of KH570-DOPO in this system. In order to overcome this drawback, the use of supercritical carbon dioxide (SCCO_2_) as a solvent has been envisaged [23,24,25]. SCCO_2_ is a commonly used supercritical fluid with moderate critical pressure and temperature (P = 7.4 MPa, Tc = 304.2 K, respectively), that is an excellent solvent for most alkoxysilanes [26]. Moreover, the SCCO_2_ properties of gas-like diffusivity and viscosity, and zero surface tension, facilitate the complete wetting of the internal surface of the mesoporous aggregates formed by nanoparticles.

In this work, KH570-DOPO was selected as a surface modifier for TiO_2_ in order to improve its compatibility with the matrix. Moreover, SCCO_2_ was used as the solvent for the effective production of TiO_2_-KH570-DOPO. The obtained nanoparticles were then incorporated into FPUF. The flame retardant performances and thermal stability of FPUF blends were characterized. The morphologies of the residual char were observed and a mode of action for the flame retardant was also proposed.

## 2. Experimental

### 2.1. Materials

TiO_2_ nanometric particles (21 nm) were supplied by Degussa (Commercial grade P25). 3-methacryloxypropyltrimethoxysilane (KH570) was obtained from Energy Chemical Co. Ltd., Shanghai, China. DOPO was supplied by H&G Chemical Co. Ltd., Jiangsu, China. Polybasic alcohol compositions (PO 715) were obtained from Shandong Ogasen New Material Co. Ltd., Shandong, China. Isocyanate (8725) was supplied by Wanhua Chemical Group Co. Ltd., Shandong, China.

### 2.2. Measurements

Fourier transform infrared (FTIR) spectra were recorded using a Bruker Tensor 27 FTIR spectrometer with 1 cm^−1^ resolution over 128 scans for KBr disk samples in the range of 4000 to 500 cm^−1^.

Thermogravimetric analysis (TGA) was carried out using a Q50 apparatus from TA Instruments at a heating rate of 10 °C/min under an air atmosphere with the temperature range from ambient temperature over 750 °C.

Transmission electron microscopy (TEM) observation was carried out with a Hitachi H-800 electron microscope. The analysis was performed at 200 kV acceleration voltage and bright field illumination under ambient temperature, and the samples were prepared by dispersing the particles into alcoholic solution with ultrasonic treatment.

Microscale combustion calorimetry (MCC) was carried out using an FAA Micro calorimeter (FAA Fire testing technology, East Grinstead, UK) operated at 1 °C/s to 750 °C in the pyrolysis zone according to ASTM D7309 method A. The combustion zone was set at 900 °C. Oxygen and nitrogen flow rates were set at 20 and 80 mL/min, respectively.

The digital photographs of calcined FPUF residues recorded using a TF-1450 thermal camera (Xiangyi instrument company, Xiangtan, China). The sample was heated to a certain temperature at the rate of 10 °C/min. Once the temperature had equilibrated at the desired point, a picture of this residue was recorded.

### 2.3. Preparation of KH570-DOPO

KH570-DOPO was prepared as previously reported [21]. Briefly, DOPO, KH570 (molar ration 1.1:1) and triethylamine were placed in a culture dish before putting them into a sealed supercritical carbon dioxide (SCCO_2_) reaction kettle. The reaction temperature was 60 °C, the pressure was 12 MPa and the reaction time was controlled for 2 h. After filtering, washing and drying, the products of KH570-DOPO were obtained.

### 2.4. Preparation of TiO_2_-KH570-DOPO

TiO_2_ (2.5 g) and KH570-DOPO (2.5 g) were dispersed into a culture dish. The same condition was used to synthesize the TiO_2_-KH570-DOPO nanoparticles. The reaction temperature, pressure and time were settled to 60 °C, 12MPa and 2 h, respectively. The precipitates were then filtered and washed several times with toluene (200 mL, six times) for ensuring that any absorbed KH570-DOPO was completely removed. The products were dried at 80 °C for 24 h, obtaining TiO_2_-KH570-DOPO. The preparation process was shown in Figure 1.

### 2.5. Preparation of FPUF/TiO_2_ Blends

Neat FPUF and the flame retardant FPUF blends were prepared by a conventional one-pot and free-rise method. Polybasic alcohol compositions and isocyanate were mixed and stirred using an electric stirrer at room temperature. Then the mixtures were poured into an aluminum cube with a dimension of 100 mm*100 mm*100 mm. After that, the FPUF blends were placed in an oven at 70 °C for 24 h in order to complete the polymerization reaction. As for the FPUF/TiO_2_-KH570-DOPO blends, a similar procedure was used and the loading of nanoparticles was settled at 1, 5 and 10 wt % respectively. These samples were noted as S0, S1, S5 and S10, correspondingly. The foam density was normalized to about 0.20 g/cm^3^ by varying the dosage of foam agent. The composites were prepared with a constant NCO index (110).

## 3. Results and Discussions

### 3.1. Characterization of TiO_2_-KH570-DOPO

#### 3.1.1. FTIR Analysis

The FTIR spectra of KH570-DOPO, TiO_2_ and TiO_2_-KH570-DOPO are shown in Figure 2. As reported, the appearance of C=O (1708 cm^−1^) from KH570 and the disappearance of P–H (2427 cm^−1^) indicate the reaction between DOPO and KH570 [27,28]. TiO_2_ exhibits a Ti–O peak at 405 cm^−1^. However, this peak shows a lower intensity in the spectrum of TiO_2_-KH570-DOPO. Moreover, the peaks from KH570-DOPO can also be found in TiO_2_-KH570-DOPO (such as C=O and C–H). It was indicated that KH570-DOPO was immobilized onto the surface of TiO_2_.

#### 3.1.2. TGA Analysis

The TGA curves of TiO_2_ and TiO_2_-KH570-DOPO under an air atmosphere are shown in Figure 3. For neat TiO_2_, rare weight loss can be observed. However, the thermal stability of TiO_2_-KH570-DOPO is significantly reduced. It shows a 53.0% residue at 750 °C because of the introduction of KH570-DOPO. Moreover, the grafting rate of KH570-DOPO on the surface of TiO_2_ can also be calculated from Figure 3. Neat TiO_2_ particles are stable throughout the test, while not for the modified TiO_2_. Therefore, the grafting level is 47.0% (100% − 53.0% = 47%), which is higher than the literature reports.

#### 3.1.3. Surface Morphology

The TEM images of TiO_2_ and TiO_2_-KH570-DOPO are shown in Figure 4. As shown in Figure 4a1,a2, neat TiO_2_ particles tend to form self-aggregations because of the abundant hydroxy groups on the surface. After the introduction of KH570-DOPO (Figure 4b1,b2), this problem is greatly restricted. As a result, smaller aggregations can be observed. Overall, it can be concluded that TiO_2_-KH570-DOPO has been successfully prepared based on the results from FTIR, TGA and TEM.

### 3.2. Characterization of FPUF

#### 3.2.1. Flame Retardancy

Microscale combustion calorimetry (MCC) is performed in order to investigate the combustion behavior of FPUF blends. The heat release rate (HRR), the temperature at which the peak heat release rate (pHRR) occurs (TP), as well as the total heat released (THR), can be obtained from this test. The HRR curves of the FPUF materials are shown in Figure 5, and other results are summarized in Table 1.

It can be seen from Figure 5 and Table 1 that the pHRR and THR values of flame retardant FPUF blends are gradually decreased with the increasing loading of TiO_2_-KH570-DOPO, and both two parameters reach their maximum values (512.9 W/g for pHRR and 26.8 kJ/g for THR) at 10% incorporation of the nanoparticles. The T_P_ values of the blends exhibits a similar tendency. A 9.4 °C delay can be found for the S10 sample. It has been reported that TiO_2_ takes effect in condensed phase by constructing dense and continuous char residues [13], while DOPO reacts in the gas phase as the radical scavenger [18]. As a result, the fire resistance of FPUR blends can be significantly enhanced.

#### 3.2.2. Thermal Stability

The TGA curves of FPUF and FPUF blends under air atmosphere are shown in Figure 6 and the key data collected from TGA curves are given in Table 2. It is suggested that neat FPUF sample exhibits a two-step degradation pathway with T_−5%_ and T_−50%_ values of 278 and 374 °C. The first step corresponds to the decomposition of the polymer main chain; the second step stands for the pyrosis of transient char. By the incorporation of nanoparticles, the FPUF blends also show a two-step degradation routine. However, the T_−5%_ and T_−50%_ values of FPUF blends are slightly increased compared with those of neat FPUF sample. The char residues of FPUF and FPUF blends after the tests are also given in Table 2. Neat FPUF sample exhibits 0.4% char residue, while the flame retardant FPUF samples show the much higher char residues. With the presence of 10% TiO_2_-KH570-DOPO, a maximum of 9.0% residues can be observed. The more the char has formed, the better is the flame retardancy. These physical barriers effectively separate the heat and oxygen from the matrix, resulting in improvement of both the fire resistance and thermal stability of FPUF blends [29].

MCC and TGA results have indicated that TiO_2_-KH570-DOPO nanoparticles can improve the flame retardancy and thermal stability of FPUF. The kinetic analysis can provide additional information to the thermal oxidative degradation of the blends. In this study, the multiple heating rate kinetics method is used to estimate the apparent activation energy by applying the Flynn-Wall-Ozawa method [30,31,32] specifically derived for heterogeneous chemical reactions under linear heating rates (2.5, 5, 7.5, and 10 °C/min). The derivation of the Flynn-Wall-Ozawa method from the first principle was presented in 1965 [30]. As shown in Figure 7, the apparent activation energies are plotted as a function of α.

There is a small difference in the initial activation energies (when α is below 15%) for the two samples. At low temperature, FPUF undergoes the radical peroxidation chain and the addition of TiO_2_-KH570-DOPO nanoparticles cannot influence the process. Moreover, activation energy of FPUF/TiO_2_-KH570-DOPO increases rapidly as the degradation proceeds (when α is below 70%), which is very different from that of FPUF. The high activation energies of FPUF/TiO_2_-KH570-DOPO blend might be due to the effect of TiO_2_ and phosphorus on catalyzing the formation of carbonaceous char. Thus, the releasing rate of degradation volatiles is significantly reduced. The thermal stability and flame retardancy are thereby improved. The above results indicate that TiO_2_-KH570-DOPO might react in the gas phase as the radical scavenger. Details of the mechanism should be elicited in future work.

#### 3.2.3. Condensed Phase Analysis

The digital photographs of FPUF and flame retardant FPUF blends under different temperatures are shown in Figure 8 and Figure 9. The corresponding area shrinkage is given in Figure 10. It is suggested that FPUF blends exhibit a much lower area decrease compared with those of neat FPUF samples, especially after 250 °C. The incorporation of TiO_2_-KH570-DOPO takes effect in both the gas and condensed phase, resulting in better stability of the S5 sample.

The SEM images of char residues of FPUF and FPUF blends after MCC tests are shown in Figure 11. The morphology of neat FPUF sample exhibits discontinuous residual chars with lots of cracks and cavities. Heat and volatiles can easily penetrate these char layers, leading to poor fire resistance. For flame retardant FPUF blends, the construction of char layers has been significantly improved, especially for the S10 sample. It is reported that the combination of silicane and TiO_2_ is beneficial for producing physical barriers. The formation of compact and continuous char can prevent heat transfer and protect the underlying polymeric substrate, leading to the enhancement of flame retardancy [33].

## 4. Conclusions

Multi-functional nanoparticles, TiO_2_-KH570-DOPO, have been successfully prepared. The grafting level (47%) of KH570-DOPO has been significantly improved by the existence of supercritical carbon dioxide. The introduction of TiO_2_-KH570-DOPO enhances the flame retardancy of FPUF. For the FPUF blend sample containing 10% TiO_2_-KH570-DOPO, the peak heat release rate was reduced from 657.0 to 519.2 W/g (21% decrease), total heat release value also decreased from 28.9 to 26.8 kJ/g. Thermal analysis results suggested that TiO_2_-KH570-DOPO can affect the char forming process, leading to the formation of condensed residues.

## Figures and Tables

**Figure 1 polymers-11-00075-f001:**
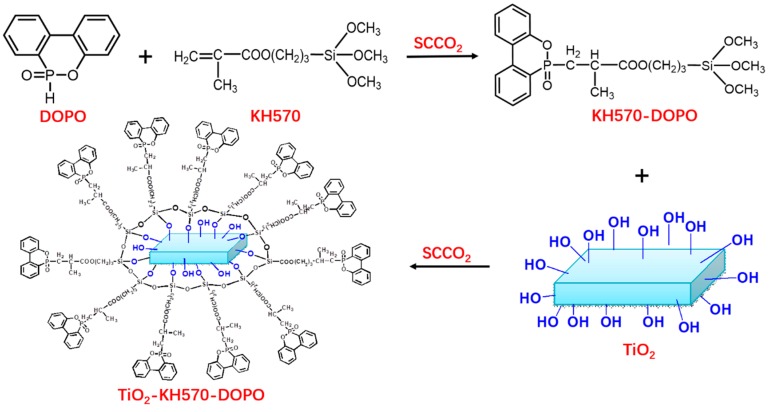
Process of TiO_2_-KH570-DOPO.

**Figure 2 polymers-11-00075-f002:**
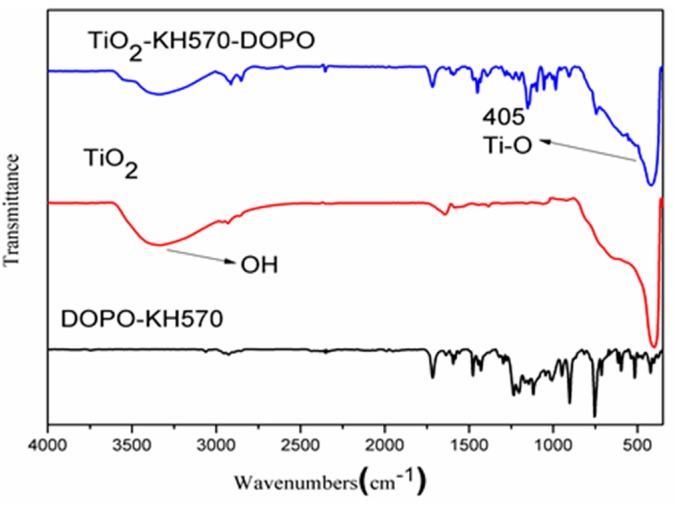
Fourier transform infrared (FTIR) spectra of TiO_2_, KH570-DOPO and TiO_2_-KH570-DOPO.

**Figure 3 polymers-11-00075-f003:**
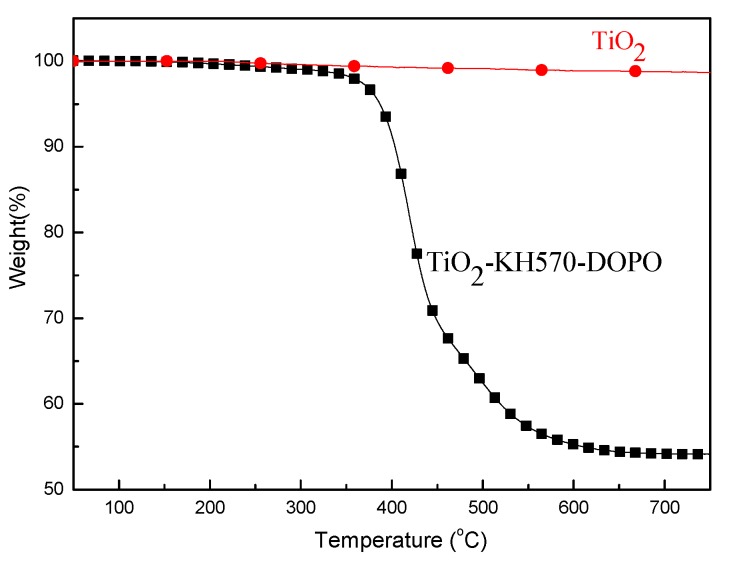
Thermogravimetric analysis (TGA) curves of TiO_2_ and TiO_2_-KH570-DOPO under air atmosphere.

**Figure 4 polymers-11-00075-f004:**
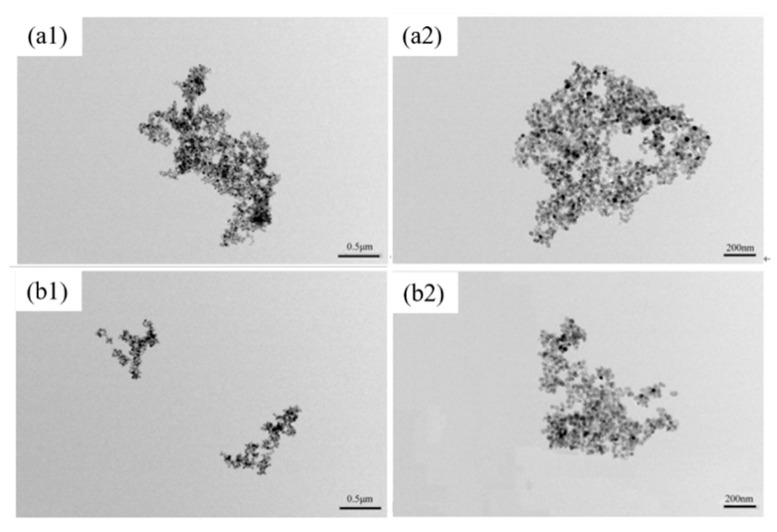
Transmission electron microscopy (TEM) image of (**a1**,**a2**) TiO_2_ and (**b1**,**b2**) TiO_2_-KH570-DOPO.

**Figure 5 polymers-11-00075-f005:**
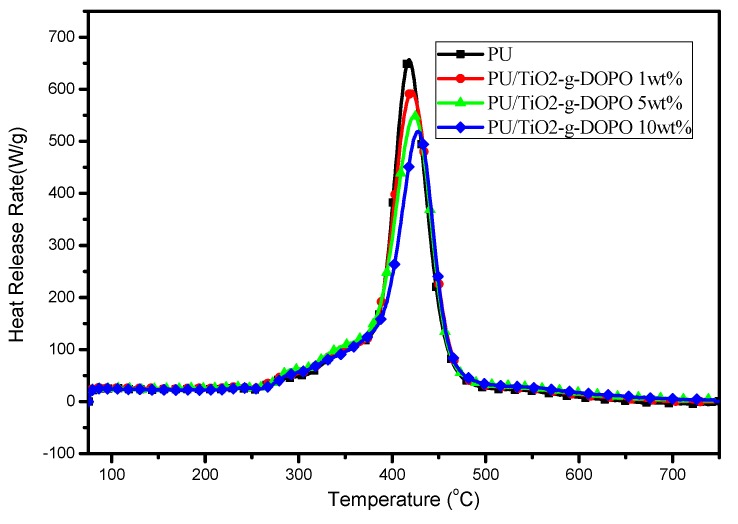
Heat release rate (HRR) curves of neat flexible polyurethane foam (FPUF) and flame retardant FPUF blends.

**Figure 6 polymers-11-00075-f006:**
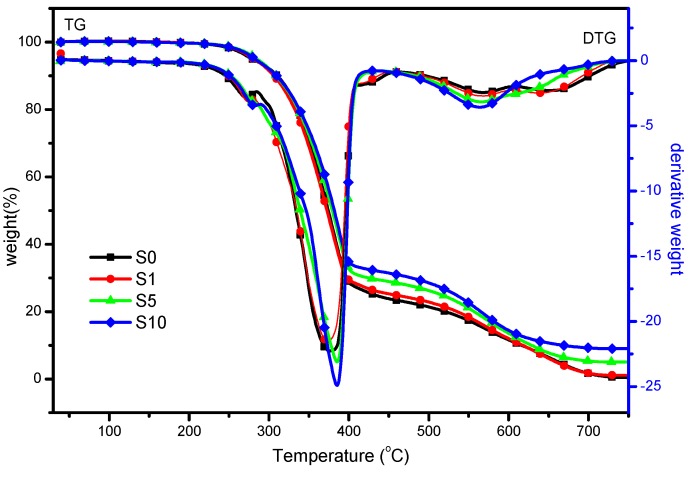
Thermogravimetric analysis (TGA) curves of neat flexible polyurethane foam (FPUF) and flame retardant FPUF blends under an air atmosphere.

**Figure 7 polymers-11-00075-f007:**
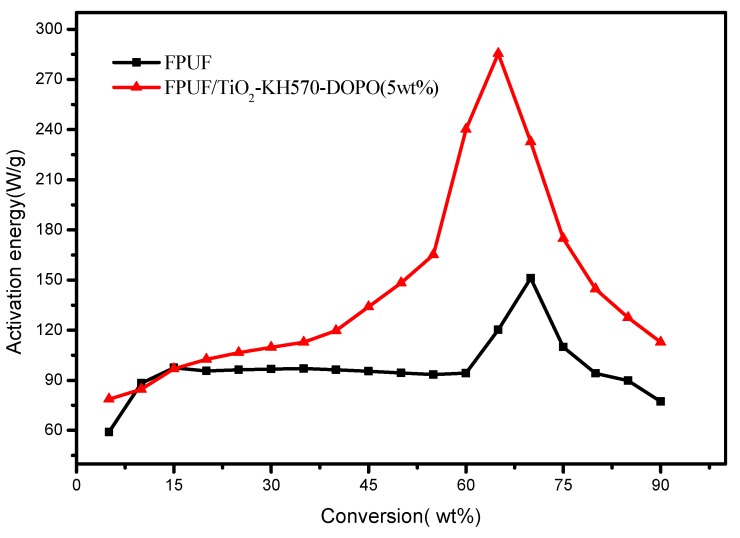
Activation energy of neat flexible polyurethane foam (FPUF) and flame retardant FPUF blends.

**Figure 8 polymers-11-00075-f008:**
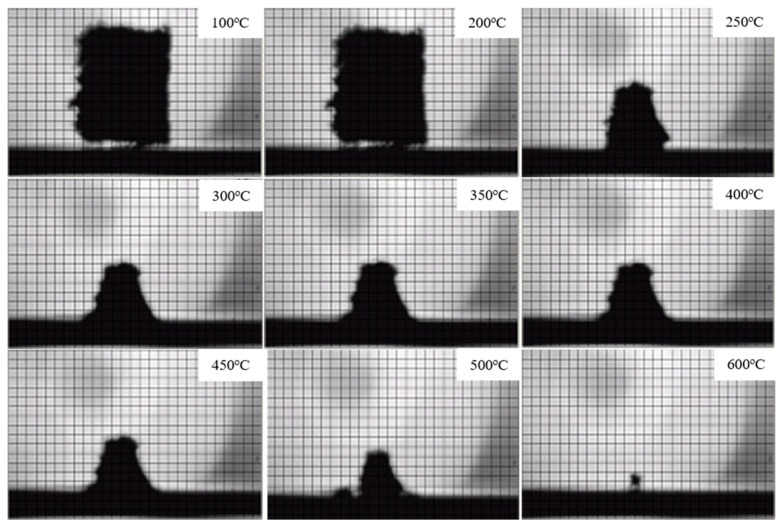
Digital photographs of neat flexible polyurethane foam (FPUF) samples at different temperatures.

**Figure 9 polymers-11-00075-f009:**
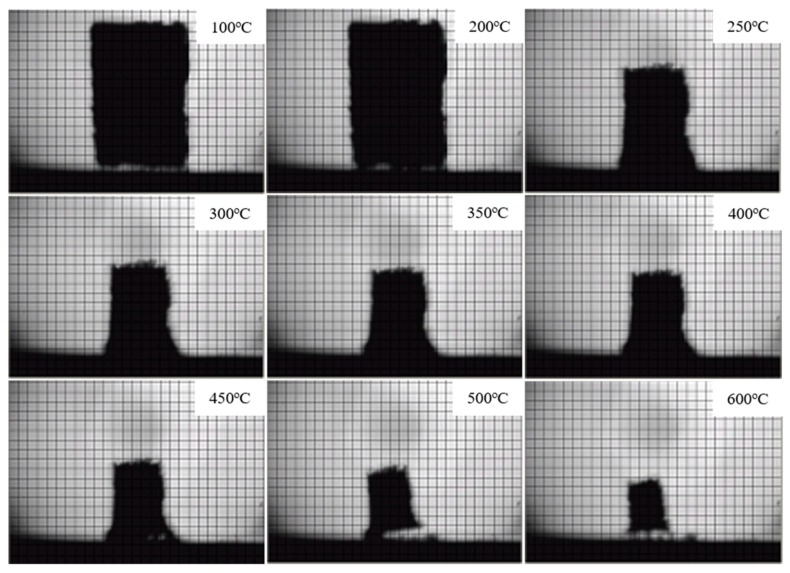
Digital photographs of flame retardant flexible polyurethane foam (FPUF) blends (S5) at different temperatures.

**Figure 10 polymers-11-00075-f010:**
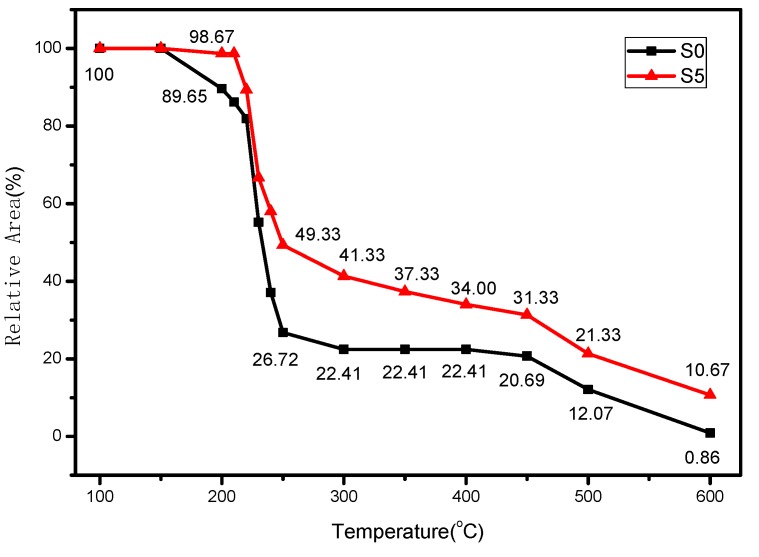
Corresponding area shrinkage for flexible polyurethane foam (FPUF) and FPUF blends.

**Figure 11 polymers-11-00075-f011:**
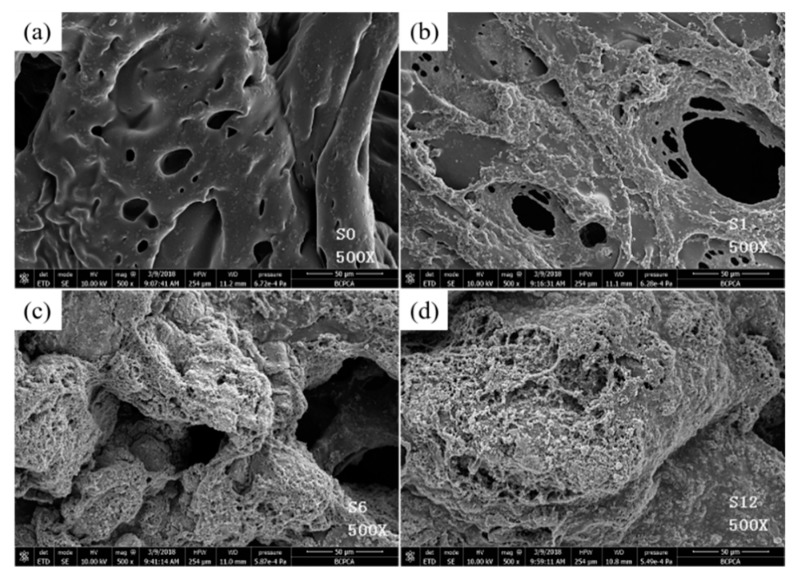
Scanning electron microscopy (SEM) images of char residues of (**a**) neat flexible polyurethane foam (FPUF), (**b**) S1, (**c**) S5 and (**d**) S10 blends after microscale combustion calorimetry (MCC) tests.

**Table 1 polymers-11-00075-t001:** The key data obtain from microscale combustion calorimetry (MCC) tests for neat flexible polyurethane foam (FPUF) and flame retardant FPUF blends.

Samples	pHRR/(W/g)	THR/(kJ/g)	T_P_/°C
S0	657.0	28.9	418.4
S1	596.5 (↓9.3%)	28.5 (↓1.4%)	421.1 (+2.7)
S5	549.7 (↓16.3%)	27.7 (↓4.2%)	423.9 (+5.5)
S10	519.2 (↓21.0%)	26.8 (↓7.3%)	427.8 (+9.4)

**Table 2 polymers-11-00075-t002:** The key data obtained from thermogravimetric analysis (TGA) tests for neat flexible polyurethane foam (FPUF) and flame retardant FPUF blends.

Samples	T_−5%_ ^a^ (°C)	T_−50%_ ^b^ (°C)	Weight Loss (wt %)		Residue(wt %)
			Step 2	Step 3	
S0	278	374	73.9	25.7	0.4
S1	284	380	72.8	26.1	2.6
S5	285	379	69.8	25.2	5.0
S10	282	381	67.3	23.7	9.0

^a^ T_−5%_: initial decomposition temperature; ^b^ T_−50%_: midpoint decomposition temperature.

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
