# Peer review of "Investigation of Flame Retardant Flexible Polyurethane Foams Containing DOPO Immobilized Titanium Dioxide Nanoparticles"

_polymers, 2019, doi:10.3390/polym11010075_

Round 1

Reviewer 1 Report

This manuscript reports the modification of the surface of titania nanoparticles with dopyl-containing triethoxysilane. A reduction in peak heat release rate for the combustion of poly(urethane) may be achieved by incorporation of the modified particles (the reduction is not huge but probably noteworthy).The level of grafting to the titania surface is somewhat uncertain. The infrared spectrum of the grafted material suggests that free hydroxyl groups remain. From TGA, the level of grafted material in the final composition is 47 wt%. How does this correspond to the number of hydroxyl groups initially present that have undergone reaction? Figure 1 is misleading since it suggests that all hydroxyl groups have been converted to adducts.

 Apparent activation energies derived from variable heating rate methods are composites reflecting contributions from all processes which change as a function of temperature (see Thermochimica Acta, 2003,396,179). The plots in Figure 7 simply suggest that thermal decomposition of poly(urethane) in the presence of additive is different from that in its absence. There is no evidence for synergism and its mention should be omitted (no experiments using the two components of the adduct as single additives in poly(urethane) have been carried out).

The manuscript will require significant rewriting. Corrections are penciled-in directly on pages of the manuscript attached. These are indicative of the kinds of changes needed throughout. In rewriting emphasis should be placed on clarity (incomplete sentences should be avoided) and readability. The compositions generated by incorporation of the modified titania  particles into poly

(urethane) are simple blends and should not be referred to as composites (no evidence for composite formation is provided). "Grafting rate" should be "grafting level". "Literature reports" should be "previously reported". Personal pronouns should be omitted. "Microscale combustion calorimetry (MCC)" is now preferred to "pyrolysis combustion flow calorimetry (PCFC)". 

Author Response

Comments and Suggestions for Authors

This manuscript reports the modification of the surface of titania nanoparticles with dopyl-containing triethoxysilane. A reduction in peak heat release rate for the combustion of poly(urethane) may be achieved by incorporation of the modified particles (the reduction is not huge but probably noteworthy).The level of grafting to the titania surface is somewhat uncertain. The infrared spectrum of the grafted material suggests that free hydroxyl groups remain. From TGA, the level of grafted material in the final composition is 47 wt%. How does this correspond to the number of hydroxyl groups initially present that have undergone reaction? Figure 1 is misleading since it suggests that all hydroxyl groups have been converted to adducts.

Response:

Thanks for your comments. Indeed, Figure 1 is misleading, and we have already revised it based on your suggestion. From FTIR analysis, the peak intensity of hydroxyl groups is decreased for TiO2-KH570-DOPO compared with that of unmodified TiO2, which directly provides the proof for the reaction between TiO2 and KH570-DOPO. However, FTIR spectra is not enough to calculate the grafting level. TGA could characterize the graft level. The precipitates were filtered and washed several times with toluene (200 mL, six times) for ensuring that any absorbed KH570-DOPO was completely removed. From the TGA results and FTIR spectrum the grafting level could be obtained. 

Apparent activation energies derived from variable heating rate methods are composites reflecting contributions from all processes which change as a function of temperature (see Thermochimica Acta, 2003,396,179). The plots in Figure 7 simply suggest that thermal decomposition of poly(urethane) in the presence of additive is different from that in its absence. There is no evidence for synergism and its mention should be omitted (no experiments using the two components of the adduct as single additives in poly(urethane) have been carried out).

Response:

Thanks for your comments. Yes, there is no evidence for synergism. That sentence was rewritten as “The high activation energies of FPUF/TiO2-KH570-DOPO blend might be owing to the effect of TiO2 and phosphorus on catalyzing the formation of carbonaceous char.” 

The manuscript will require significant rewriting. Corrections are penciled-in directly on pages of the manuscript attached. These are indicative of the kinds of changes needed throughout. In rewriting emphasis should be placed on clarity (incomplete sentences should be avoided) and readability. The compositions generated by incorporation of the modified titania particles into poly(urethane) are simple blends and should not be referred to as composites (no evidence for composite formation is provided). "Grafting rate" should be "grafting level". "Literature reports" should be "previously reported". Personal pronouns should be omitted. "Microscale combustion calorimetry (MCC)" is now preferred to "pyrolysis combustion flow calorimetry (PCFC)".

Response:

Thanks for your comments. We have revised our manuscript accordingly. The PCFC has changed into MCC, “Grafting rate” has corrected into “Grafting level”, FPUF composites has also been revised into FPUF blends. The English has been carefully checked and revised. Detailed information please see the revised manuscript.

Reviewer 2 Report

First of all, English has to be slightly improved, but these are not very serious issues. 

Next, in case of foams, values of apparent density always should be presented. Incorporation of solid particles always influences the viscosity of reaction mixture during polymerization, hence affects the density and cellular structure of foams. And, as commonly known, these parameters has influence on flammability of materials. Therefore, Authors should present value of apparent density. Also, since Authors are already using SEM, images of foams should be also presented, in order to observe the changes of cellular structure related to the incorporation of flame retardant particles. 

Author Response

First of all, English has to be slightly improved, but these are not very serious issues.

Response:

Thanks for your comments. The English has been carefully checked and revised. Detailed information please see the revised manuscript.

Next, in case of foams, values of apparent density always should be presented. Incorporation of solid particles always influences the viscosity of reaction mixture during polymerization, hence affects the density and cellular structure of foams. And, as commonly known, these parameters has influence on flammability of materials. Therefore, Authors should present value of apparent density. Also, since Authors are already using SEM, images of foams should be also presented, in order to observe the changes of cellular structure related to the incorporation of flame retardant particles.

Response:

Thanks for the comment. The density of foam could influence the property of FPUF. In this study, the foam density was normalized to about 0.20 g/cm3 by varying the dosage of foam agent. The composites were prepared with a constant NCO index (110).

          We characterized the cellular structure of the foam as shown in Figure S1. It is known that FPUF has open cell structure. As shown in the images below, the foam structure did not significantly change. I think it is not necessary to present the picture in the manuscript.

Figure S1. Cellular structure of the foams.

Round 2

Reviewer 2 Report

Everything in order after corrections.